# Measuring Systematic Generalization in Neural Proof Generation with Transformers

**Nicolas Gontier**
Quebec Artifical Intelligence Institute (Mila),
Polytechnique Montréal
gontiern@mila.quebec

**Koustuv Sinha**
Quebec Artifical Intelligence Institute (Mila),
McGill University
Facebook AI Research

**Siva Reddy**
Quebec Artifical Intelligence Institute (Mila),
McGill University
Facebook CIFAR AI Chair

**Christopher Pal**
Quebec Artifical Intelligence Institute (Mila),
Polytechnique Montréal
ElementAI
Canada CIFAR AI Chair

## Abstract

We are interested in understanding how well Transformer language models (TLMs) can perform reasoning tasks when trained on knowledge encoded in the form of natural language. We investigate their systematic generalization abilities on a logical reasoning task in natural language, which involves reasoning over relationships between entities grounded in first-order logical proofs. Specifically, we perform soft theorem-proving by leveraging TLMs to generate natural language proofs. We test the generated proofs for logical consistency, along with the accuracy of the final inference. We observe length-generalization issues when evaluated on longer-than-trained sequences. However, we observe TLMs improve their generalization performance after being exposed to longer, exhaustive proofs. In addition, we discover that TLMs are able to generalize better using backward-chaining proofs compared to their forward-chaining counterparts, while they find it easier to generate forward chaining proofs. We observe that models that are not trained to generate proofs are better at generalizing to problems based on longer proofs. This suggests that Transformers have efficient internal reasoning strategies that are harder to interpret. These results highlight the systematic generalization behavior of TLMs in the context of logical reasoning, and we believe this work motivates deeper inspection of their underlying reasoning strategies.

## 1   Introduction

Systematic Generalization has been characterized as the capacity to understand and produce a potentially infinite number of novel combinations from known components (Chomsky, 1957; Montague, 1970). For example, in Figure 1, a model could be exposed to a set of facts (e.g., "*Nat is the granddaughter of Betty*", "*Greg is the brother of Nat*", "*Flo is the sister of Greg*"), but not to all the possible facts that can be inferred by combination of the known components (e.g., "*Flo is the granddaughter of Betty*"). More recent work has examined systematic generalization in terms of the ability of "a model to manipulate concepts in new combinations after being trained on all concepts, but only on a limited set of their combinations" (Bahdanau et al., 2019a). This view of systematic generalization shifts emphasis from reasoning to learning. Here we examine systematic generalization through measuring the ability of a model to reason about new inference step combinations despite being trained on a limited subset of them.

Recent developments in natural language processing (NLP) have shown that Transformer (Vaswani et al., 2017) Language Models (TLMs) are able to capture linguistic knowledge (Peters et al., 2018; Goldberg, 2019; Tenney et al., 2019), and yield state-of-the-art performance for many NLP tasks (Radford et al., 2018; Devlin et al., 2019), including but not limited to answering reading comprehension questions (Radford et al., 2019; Brown et al., 2020) and generating factual knowledge (Petroni et al., 2019) with little to no task supervision. These models are optimized on large corpora to predict the next words or a set of masked words in a sentence. While yielding impressive results, it is not clear if TLMs rely on many superficial patterns in the data or if they actually learn re-usable skills, enabling them to generalize to new tasks by leveraging the compositionality of those skills (Lake and Baroni, 2018; Liška et al., 2018). Training on massive data can give certain advantages with respect to understanding the meanings of words, but we conjecture that such data gives models less experience with reasoning over inference chains.

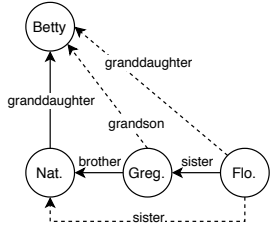

Figure 1: Example of a CLUTRR graph with known facts (solid lines) and unknown facts to infer (dotted lines).

In our work, we study the less understood issues related to how well TLMs are able to perform long chains of reasoning. In particular, we use TLMs for the task of theorem proving, where facts and proofs are specified in natural language. Using theorem proving, we test if TLMs can generate interpretable proofs with logically consistent language modeling as their main objective. In particular, we study their *behavior* as logical reasoners on text by analyzing the generated proofs and the final answer. This setup allows us to evaluate the reasoning and generalization capabilities of TLMs. Recent work such as Petroni et al. (2019); Raffel et al. (2020); Brown et al. (2020) suggest that language models can be treated as knowledge bases. This directly motivates us to investigate if language models can also learn certain reasoning strategies. Studying these abilities can give us insights to facilitate the use of such models as dynamic knowledge bases that could infer new knowledge even if it is not seen during pre-training.

For natural language theorem proving, we use the question answering CLUTRR benchmark suite (Sinha et al., 2019) to perform controlled studies. This dataset is of interest because: (i) the compositional nature of tasks involved make it well suited for evaluating systematic generalization, and (ii) each question–answer pair is accompanied by a proof that can be used to explain how to arrive at the answer. We use this dataset as a medium to understand the reasoning capacity of TLMs.

Our experiments reveal the following:

1. TLMs suffer from length generalization: they cannot *extrapolate* to proofs requiring more proof steps than seen during training time.
2. They generalize better when trained to generate long proofs compared to short proofs.
3. They generalize better when trained to generate backward-chaining proofs rather than forward-chaining.
4. Surprisingly, they generalize better when they are trained to directly generate the answer instead of learning to generate the proof and then the answer.

To the best of our knowledge, we are the first to use a language modeling objective to do interpretable theorem proving with a Transformer. We hope that this work can shed some light on the reasoning capacity of TLMs and inspire future research to design models with greater reasoning capacity.

## 2   Related Work

Systematic generalization has recently been in spotlight due to its importance in understanding the strengths and weaknesses of neural networks. Bahdanau et al. (2019a,b) identify and evaluate the generalization capacity of visual question answering models. We however focus this study on a fully natural language domain. Dasgupta et al. (2019) introduce a natural language inference (NLI) dataset which proves to be challenging for language understanding models for compositional generalization. Goodwin et al. (2020) also evaluate systematic generalization in an NLI setting with controlled test cases to observe the failures of neural architectures. We however focus this study on the systematic generation of logical reasoning sentences by Transformer-based (Vaswani et al., 2017) language models in a question answering setting with the CLUTRR suite (Sinha et al., 2019). Similar datasets include SCAN (Lake and Baroni, 2018) which has been instrumental to test systematic generalization

| | raw | facts | amt |
|---|---|---|---|
| **story** | [(Natasha, granddaughter, Betty), (Florence, sister, Gregorio), (Gregorio, brother, Natasha)] | \<STORY\><br>Natasha is a granddaughter to Betty.<br>Florence is Gregorio 's sister.<br>Gregorio is a brother of Natasha. | \<STORY\> Betty likes picking berries with<br>her son 's daughter. Her name is Natasha.<br>Gregorio took his sister, Florence, to a baseball game.<br>Gregorio and his sister Natasha love it when their<br>grandmother visits because she spoils them.<br>She is coming this week to watch them while<br>their parents are out of town. |
| **query** | (Florence, _, Betty) | \<QUERY\> Who is Florence for Betty ? | |
| **proof** | [{(Florence, granddaughter, Betty): [(Florence, sister, Gregorio), (Gregorio, grandson, Betty)]}, {(Gregorio, grandson, Betty): [(Gregorio, brother, Natasha), (Natasha, granddaughter, Betty)]}] | \<PROOF\><br>since Florence is a sister of Gregorio, and Gregorio is a grandson to Betty,<br>then Florence is a granddaughter to Betty.<br>since Gregorio is a brother of Natasha, and Natasha is the granddaughter of Betty,<br>then Gregorio is a grandson of Betty. | |
| **answer** | granddaughter | \<ANSWER\> Florence is the granddaughter of Betty | |

Table 1: CLUTRR example of level 3 (ie: 4 entities, 3 relations, 2 proof steps). The proof follows the `short-proof-rev` strategy. We refer the reader to Figure 1 to visualize the corresponding graph in which solid lines refer to the facts given in the story and dotted lines refer to the new facts inferred in each proof step.

(Lake, 2019; Baroni, 2020) and CFQ (Keysers et al., 2020) which measures the systematicity of language understanding via a question answering setup. Sinha et al. (2019) propose a series of baseline models with the CLUTRR dataset but none of them took advantage of the provided proof attached with each example. In addition, their Transformer baselines were not fine-tuned on the task. Unlike them, we focus on learning and generating proofs for studying systematic generalization.

Neural proof generation (Sekiyama et al., 2017) and neural theorem proving (Rocktäschel and Riedel, 2017; Weber et al., 2019; Minervini et al., 2020) have been explored in previous work. They tend to combine symbolic and statistical approaches to leverage the compositionality and interpretability of symbolic systems and the flexibility of statistical systems. Nevertheless, these combined systems all assume some predefined set of atoms and rules making up the environment. We instead use natural language text to define our environment and measure the limits of a purely statistical approach.

Similarly to us, Clark et al. (2020) leverage logical rules expressed in natural language to answer compositional questions. However their system is not generative, rather they predict a true/false binary label on candidate answers. We instead focus on the systematic generalization capacity of generating proofs and using them to generate the final answer.

## 3 Evaluating systematic generalization through interpretable reasoning

### 3.1 The task

**Background**. We use the family relation CLUTRR benchmark suite (Sinha et al., 2019) to generate our dataset[1]. Each example is composed of: (i) a family graph $G = (V, E)$ (referred as *story*) with entities as nodes ($v \in V$) and relationships as edges ($e \in E$), (ii) a *query* about the relationship between two entities ($v_1, \_, v_n$) separated by more than one hop in the family graph (iii) a reasoning path (referred as *proof*) expressed as a list of $(v_i, e_j, v_k)$ tuples, referred to as *facts* and (iv) the target relationship $e^*$ between the two queried entities (referred to as the *answer*). The dataset contains 272 distinct entities and 20 relationship types, ordering to $\sim$ 1.5M possible facts. Each $(v_i, e_j, v_k)$ fact can be expressed in natural language using either one of 5 factual sentences (referred to as `facts` template), or by using one of $6,000$ noisy but more natural sentences written by mechanical turkers (refered as `amt` template). Family graphs are expressed using either the `facts` template or the `amt` template, while queries, proofs and answers are always expressed with the `facts` template. A CLUTRR example can be seen in Table 1 and Figure 1.

**Terminology**. In order to evaluate systematic generalization, we define the following building blocks that constitute a *proof*:

- `entity`: one node (e.g., *"Anna"*).
- `relation`: one edge (e.g., *"mother"*).
- `fact`: one factual sentence representing a $(v_i, e_j, v_k)$ tuple using `facts` template (e.g., *"Anna is the mother of Bob"*).

| | |
|---|---|
| **sp** | since Gregorio is a brother of Natasha, and Natasha is the granddaughter of Betty, then Gregorio is a grandson of Betty.<br>since Florence is a sister of Gregorio, and Gregorio is a grandson to Betty, then Florence is a granddaughter to Betty. |
| **spr** | since Florence is a sister of Gregorio, and Gregorio is a grandson to Betty, then Florence is a granddaughter to Betty.<br>since Gregorio is a brother of Natasha, and Natasha is the granddaughter of Betty, then Gregorio is a grandson of Betty. |
| **lp** | since Gregorio is the brother of Natasha, and Natasha is the granddaughter of Betty, then Gregorio is the grandson of Betty.<br>since Florence is the sister of Gregorio, and Gregorio is the brother of Natasha, then Florence is the sister of Natasha.<br>since Florence is the sister of Natasha, and Natasha is the granddaughter of Betty, then Florence is the granddaughter of Betty. |
| **lpr** | since Florence is the sister of Natasha, and Natasha is the granddaughter of Betty, then Florence is the granddaughter of Betty.<br>since Florence is the sister of Gregorio, and Gregorio is the brother of Natasha, then Florence is the sister of Natasha.<br>since Gregorio is the brother of Natasha, and Natasha is the granddaughter of Betty, then Gregorio is the grandson of Betty. |

Table 2: Proof resolution types for an example of level 3. We refer the reader to Figure 1 for the kinship graph corresponding to this example. **sp**=short-proof, **spr**=short-proof-reversed, **lp**=long-proof, **lpr**=long-proof-reversed.

- `proof_step`: one inference step combining two `facts` to get a new one (e.g., *"since Anna is the mother of Bob and Bob is the brother of Carl then Anna is the mother of Carl."*).
- `proof`: the entire *resolution chain*, consisting of multiple ordered `proof_steps`.

Following the setup of CLUTRR, we define the relative *difficulty* of individual examples according to the number of edges present in the family graph. For instance, Table 1 and Figure 1 show a level-3 example because there are 3 solid edges (known facts) between 4 entities. In general, a **level $k$ task consists of $k$ edges** (corresponding to $k$ sentences in the story) **between $k + 1$ nodes and $k − 1$ hidden edges to infer** (corresponding to $k − 1$ proof steps to solve the task). As the levels increase, so does the number of sentences in the story and the number of proof steps in the proof.

**Problem Setup**. We trigger a model to: (1) given a story and query, generate a proof followed by an answer, and (2) given a story, query, and a proof, generate an answer. In particular, we train a Transformer-based decoder (Liu et al., 2018) with the language modeling objective on entire sequences of "<STORY> [story] <QUERY> [query] <PROOF> [proof] <ANSWER> [answer]":

$$L(\theta) = \sum_i \log P(w_i | w_1, \ldots, w_{i-1}; \theta)$$

This setup enables to generate both the answer to a query and the proof to arrive at this answer, given as input the family graph story and a question. Concretely, we inject sequences of the story and query having delimiters "<STORY>" and "<QUERY>" to the language model and trigger it to generate the corresponding proof and answer with tokens "<PROOF>" and "<ANSWER>" respectively.

## 3.2 Proof resolution strategies

In our task, we turn language models into approximate proof generators. Specifically, we train TLMs to generate `proofs` (as defined in Section 3.1). We do not explicitly perform inference on the generated `proofs`, but reformulate the language generation objective to generate the inferred answer after the `proof` sequence. This allows to leverage TLMs to generate *forward* and *backward* chaining resolution paths used in Inductive Logic Programming (ILP) (Evans and Grefenstette, 2018). In our case, these resolution paths are expressed in natural language. To simulate approximate theorem generation, we introduce four different types of `proof` that can be used to derive the answer given a story and query. An example of each type can be seen in Table 2 and we describe them below:

| ANON TEST | lvl.**2** | lvl.**3** | lvl.**4** | lvl.**5** | lvl.**6** | lvl.**7** | lvl.**8** | lvl.**9** | lvl.**10** |
|---|---|---|---|---|---|---|---|---|---|
| **proofs** (*many proof steps*) | 16.28% | 0% | 0% | 0% | 0% | 0% | 0% | 0% | 0% |
| **proof steps** ("*since A-r1-B and B-r2-C then A-r3-C*") | 73.08% | 58.06% | 52.75% | 54.28% | 50.93% | 59.04% | 56.92% | 53.55% | 52.17% |
| **facts** (*A-r-B*) | 100% | 100% | 100% | 100% | 100% | 100% | 100% | 100% | 100% |
| **entities** (*A*) | 100% | 100% | 100% | 100% | 100% | 100% | 100% | 100% | 100% |
| **relations** (*r*) | 100% | 100% | 100% | 100% | 100% | 100% | 100% | 100% | 100% |

Table 3: Percentage of the test proof's building blocks also present in the training set (composed of levels 2, 4, 6) for all levels. We colored all cells with a value of 100% to better visualize which building blocks were entirely contained in the training set.

**short-proof-rev (spr)**. This setup is the backward-chaining resolution path provided by the CLUTRR dataset, which is generated by recursive application of the kinship logical rules. This proof strategy can be viewed as an *explain-why* scenario, where the first sentence in the proof contains the answer (target relationship) and the subsequent sentences contain the intermediate steps required to explain that answer. We refer the reader to Sinha et al. (2019) for further details on the generation of this proof setting.

**short-proof (sp)**. Here we reverse the resolution chain provided by the CLUTRR dataset by swapping all sentences from the `short-proof-rev` setup. Doing so, we arrive at a forward-chaining inference path, in which the final proof step consists of the target relationship. Specifically, the first sentence in the proof combines two facts from the given story to infer a new fact. In subsequent proof steps, the inferred fact from the previous step is combined with a fact from the story, to infer a new fact until the answer is found.

**long-proof (lp)**. Forward-chaining inference in ILP consists of generating all possible new facts from the starting facts, and evaluate each of them for the resolution of the target answer (Russell and Norvig, 2010). Similarly, in this setup, we extend the `short-proof` setup where we attempt to infer all possible facts given the ones present in the input story. Each proof step combines any two previously known facts to infer a new fact until the answer is found. Pseudo-code for generating this type of proof can be found in Appendix 6.2.

**long-proof-rev (lpr)**. This setting is the same as the previous one, but in reverse. It starts from the answer and goes back to the facts originally given in the story. This resolution strategy can be viewed as a backward-chaining strategy where all possible paths are considered. This proof strategy is obtained by swapping all sentences from the `long-proof` setting.

We compare each strategy in our experiments to understand which form of logical resolution is easier to learn for TLMs. In particular, we note that the reversed proof strategies (`spr` and `lpr`) fall in the backward-chaining family of logical resolution, while the non-reversed strategies (`sp` and `lp`) represent the forward-chaining resolutions. Backward-chaining family features the proof step containing the answer at the beginning of the proof. On the other hand, forward-chaining type proofs (`sp` and `lp`) feature the proof step containing the answer at the end of the proof.

### 3.3 Systematic generalization in proof generation

Now that we have defined the task and various proof generation strategies available in our setup, we proceed to define the aspects of generalization we aim to test. Our initial CLUTRR formulation tested the generalization capacity of a model to new `facts` hence new `proof_steps` and new `proofs`, after being trained on all `entities` and `relations`. Initial experiments on this setup showed that TLMs fail to generalize to unseen `facts`. Due to the presence of a large number of entities in CLUTRR, we ended up with combinatorially large number of possible facts. The model may thus not be able to learn how to represent each entity effectively, hence reducing its chances to learn higher-order structures such as unseen `facts`. Experimental results on this original setting are provided in Appendix 6.1.

We instead slightly simplify the generalization evaluation and allow the model to also be exposed to all possible `facts`. This formulation tests a model capacity to generalize to new `proof_steps` hence new `proofs`, after being trained on all `entities`, `relations` and `facts`. Since providing a training corpus covering all possible facts would significantly increase the training data, we instead reduce

the number of entities by replacing them by one of $k^2$ randomly sampled entity tokens, resulting in significantly fewer possible facts, and thus all facts being contained in the training set (Table 3).

**Interpolation and Extrapolation**. Having access to the level of difficulty of each test examples, we evaluate both how Transformers can generalize to *unseen* proofs of the *same* difficulty as seen during training (inductive generalization); and how they can generalize to *unseen* proofs of *unseen* difficulty levels. In particular, we test *interpolation* in which the testing difficulty levels less than training levels; and *extrapolation* in which the test difficulty levels are higher than training levels. This systematically tests the length generalization capabilities of TLMs in logical theorem proving.

## 4   Experiments and Analysis

We aim to answer the following questions to analyze the proof generation capabilities of Transformer-based language models (TLMs):

1. Are TLMs able to reason better after being trained to generate interpretable proofs expressed in natural language?
2. Which types of proof are easier to learn and to generate for TLMs?
3. Which types of proof are more useful for TLMs to generate accurate answers?

**Setup**. In all our experiments we used a Transformer decoder architecture (Liu et al., 2018) with 2.5M and 3.5M parameters with a vocabulary size of 90 and $1,800$ tokens for stories expressed with the `facts` and `amt` template respectively. Detailed parameter settings for our models are given in Appendix 6.3. We also ran preliminary experiments with a larger model (145M parameters) (Appendix 6.4), with a GPT2 model (Radford et al., 2019) (Appendix 6.5), and with a more complex network (an encoder-decoder transformer) (Appendix 6.6) but found similar conclusions or further investigation being required. We generate $390,000$ CLUTRR examples of level 2 to 10. We train the models on $300,000$ examples of levels $2, 4$ and $6$ and evaluate the model on a test set of $10,000$ examples for all levels from 2 to 10. Specifically, we test levels 3 and 5 for interpolation, levels $2, 4$ and 6 for inductive generalization and levels $7, 8, 9$ and 10 for extrapolation.

**Evaluation Metrics**. In the following experiments, we evaluate both the generated *proof factual consistency* (that we denote '*validity*' in the rest of this document) and *answer accuracy*. The answer is defined as the first sentence after the "<ANSWER>" tag in the generated sequence. Since all answers during training were expressed using the `facts` template, we inverse this template to extract the $(entity, relation, entity)$ triple from the generated answer. If the extraction fails, we consider the generated answer wrong. We then compare the extracted triple to the ground truth provided in the CLUTRR dataset. For comparison, in all experiments, we also report the accuracy of the naive most-frequent-relation (MFR) baseline consisting of predicting the relation that is the most frequent in the training set for the queried entity pair.

A proof is defined as the ordered sequence of all sentences generated between the "<PROOF>" and "<ANSWER>" tokens. For validating a proof, since all proofs during training were expressed using the `facts` template, we inverse this template to extract all $(entity, relation, entity)$ triples from the generated proof sentences. If the extraction process fails at any point, the entire proof is considered invalid. The ordered sequence of each proof step is then evaluated against the transitivity rules defined by the CLUTRR environment. In addition, we also check that all the facts necessary for the proof are either given in the input story, or inferred from a previous proof step. If any of these conditions fail, we consider the proof invalid, otherwise we consider the proof 'valid' (ie: factually consistent).

**No proof setup**. In addition to the four proof strategies defined in Section 3.2, we also compare in all our experiments with a model that is trained to directly generate the answer after the story and query. In particular, this *no-proof* model is trained on sequences of "<STORY> [story] <QUERY> [query] <PROOF> none . <ANSWER> [answer]". This allows us to estimate how important is the proof for our models to be able to generalize.

### 4.1   Answer Accuracy

We evaluate the answer accuracy of models trained with different proof settings on the test set described earlier by Table 3. Each model is given as input a story, query and the proof trigger token

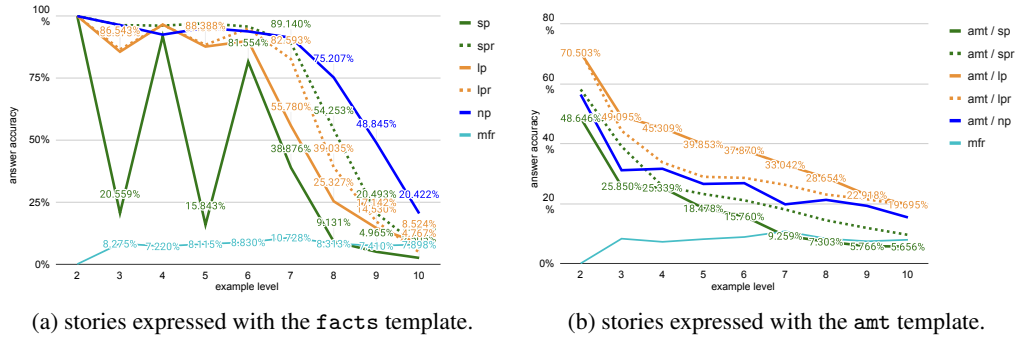

(a) stories expressed with the `facts` template.          (b) stories expressed with the `amt` template.

Figure 2: Answer accuracy for all test levels from 2 to 10. The models are given as input "<STORY> [story] <QUERY> [query] <PROOF>" and they generate the proof and answer. The models are trained on levels 2, 4, 6 only. Different proof settings are evaluated: **sp**=short-proof, **spr**=short-proof-reversed, **lp**=long-proof, **lpr**=long-proof-reversed, **np**=no-proof. We also report the naive most-frequent-relation (**mfr**) baseline.

("<STORY> [story] <QUERY> [query] <PROOF>"), and we let them decode the next tokens, that is, the proof followed by the answer.

**Q**: *Are TLMs able to generalize to unseen proof steps?* **A**: *For simple language, yes in interpolation and no in extrapolation. For complex language, no in both cases.*

In Figure 2a we evaluate models trained wit stories expressed with the `facts` template. We observe that in all proof setups, with the exception of short-proofs, TLMs are able to systematically generalize to predict the correct answer inferred from unseen `proof_steps` and `proofs`, both in inductive (levels 2, 4, 6) and interpolation levels (levels 3 and 5). However, in all proof setups TLMs have difficulties to extrapolate to longer problems requiring a larger number of reasoning steps, conforming to length generalization issues discovered in related tasks (Lake, 2019).

In Figure 2b we note that models trained on noisy `amt` stories fail to systematically generalize to predict the correct answer. In addition, we can see a linear decrease in accuracy with the level of difficulty. Having to de-noise the input stories to extract relevant kinship relations, in addition to running logical inference, makes the task much more challenging for our network. We conjecture that generalizing in this harder setting may require additional capacity added to the model, either in terms of model size, model architecture, training data, or a combination of all the above. For instance, we explore the benefit of fine-tuning GPT2 (Radford et al., 2019) in Section 6.5 as an initial step, but leave room for further improvement in future work.

**Q**: *Which reasoning strategy generalizes better?* **A**: *Backward-chaining is better than forward-chaining, but no-proof can be better than both. Long-proofs are better than short-proofs.*

We observe that backward proof strategies (spr, lpr) better help the model to answer accurately than their respective forward strategies (sp, lp) (Figure 2), with the exception of long proofs in the `amt` story template. This suggests that backward chaining is easier to learn, easier to use, or both, than forward chaining for TLMs. We believe this effect is due to the position-dependent exploitation of TLMs. Indeed, the answer is in the first generated proof-step in case of backward-chaining proofs. In addition, we note in Figure 2 that long-proofs (lp, lpr) yield better generalization performance than short-proofs (sp, spr) with the exception of reversed strategies in the `facts` story template.

It is also interesting to see that models trained to go directly to the answer by generating the "*none*" token as a proof tend to perform better than all other models required to generate the proof in `facts` stories (Figure 2a). One hypothesis is that the generated proof may be invalid most of the time and hence the extra information given by the proof is actually deteriorating the model's performance. To see if that may be the case, we next look at the validity of the generated proofs for all models (except the trivial no-proof).

## 4.2 Proof Validity

We evaluate the proof validity of models trained with different proof settings on the test set (previously described by Table 3) in Figure 3. Similarly as above, each model is given as input a story and query

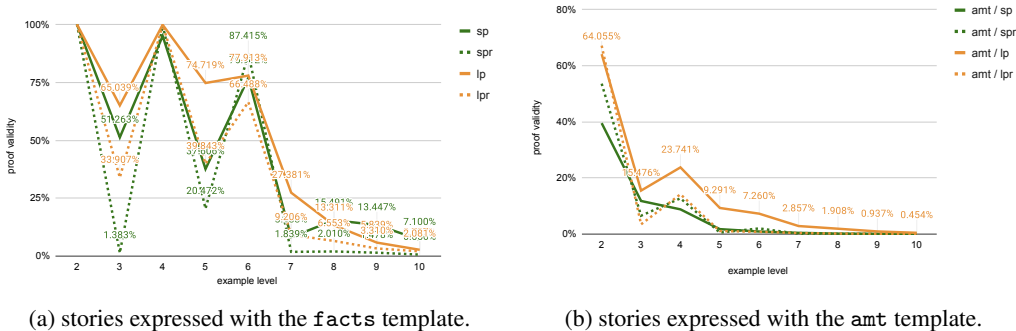

(a) stories expressed with the `facts` template.          (b) stories expressed with the `amt` template.

Figure 3: Proof validity for all test levels from 2 to 10. The models are given as input "<STORY> [story] <QUERY> [query] <PROOF>" and they generate the proof and answer. The models are trained on levels 2, 4, 6 only. Different proof settings are evaluated: **sp**=short-proof, **spr**=short-proof-reversed, **lp**=long-proof, **lpr**=long-proof-reversed, **np**=no-proof.

and we trigger the model to decode the proof and answer with the trigger tokens "<PROOF>" and "<ANSWER>" respectively.

**Q**: *Which reasoning strategy is easier to generate?* **A**: *forward-chaining is easier than backward-chaining and long-proofs are easier than short-proofs.*

From Figure 3a we observe that forward-chaining strategies (sp, lp) tend to be easier to generate than their respective reversed strategies (spr, lpr). This is contrary to the previous observation where backward-chaining strategies were easier for the models to understand. We believe that this is due to the fact that the model has a higher chance of generating the first proof step correctly than the final proof step. Since backward chaining proofs contain the answer in the first proof step, when re-using that information to predict the answer, there is a higher chance that the answer will be correct. This explains why the answer accuracy of such model is relatively high while their proof validity is relatively low.

In addition, we observe that in both `facts` and `amt` stories (Figure 3), long proof strategies are easier to generate than shorter ones. This was not expected at first since long sequences are usually harder to model in language models. One hypothesis is that since long-proofs come from a systematic construction (see Appendix 6.2) they are easier to generate than the more arbitrary short proofs.

**Q**: *Are TLMs able to generate valid proofs of unseen lengths?* **A**: *No.*

We observe that valid proofs are difficult to generate for TLMs in unseen difficulty levels, both in interpolation and extrapolation setting (Figure 3a). This partially explains why the no-proof setting in the previous section yielded better generalization performances. In addition, we note in Figure3b that the generated proofs from models trained on noisy `amt` stories are mostly invalid. We believe that this is due to the fact that models need to de-noise the information from the input story in addition to generating a valid proof, making the task much harder. To understand if models rely on the validity of the proof, we next evaluate their answer accuracy when given the real proof as input rather than the generated one.

## 4.3   Proof is given

To understand if models rely on the proof, we again evaluate the answer accuracy as in Section 4.1, but this time the models are given as input the story, the query and the real proof followed by the answer trigger token: "<STORY> [story] <QUERY> [query] <PROOF> [proof] <ANSWER>". We then let the language model decode the next tokens making up the answer. Note that the no-proof model is given "none" as its "[proof]" so we don't expect this model performance to change from Section 4.1.

**Q**: *Are ground-truth proofs useful for TLMs to generalize systematically?* **A**: *Yes.*

When the proof is provided in the input, all models outperform the no-proof model in inductive and interpolation test cases (Figure 4). In extrapolation test cases, models trained on `facts` stories (Figure 4a) benefit from the proof compared to Section 4.1, and models trained with `amt` stories outperform the no-proof model (Figure 4b). This suggests that models do learn to use the correct

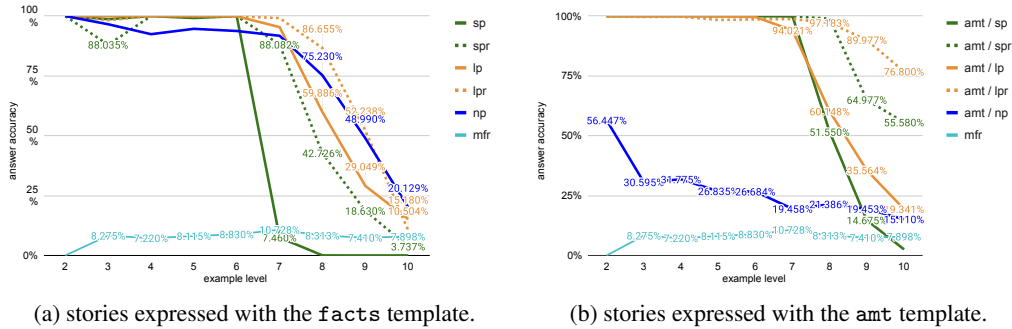

(a) stories expressed with the `facts` template.　　(b) stories expressed with the `amt` template.

Figure 4: Answer accuracy for all levels from 2 to 10. The models are given as input "<STORY> [story] <QUERY> [query] <PROOF> [proof] <ANSWER>" and they generate the answer. The models are trained on levels 2, 4, 6 only. Different proof settings are evaluated: **sp**=short-proof, **spr**=short-proof-reversed, **lp**=long-proof, **lpr**=long-proof-reversed, **np**=no-proof. We also report the naive most-frequent-relation (**mfr**) baseline.

proof to better generalize during inference. However, as the difficulty of the examples increases, the generalization performance of all models decreases. Even when given the proof containing the correct answer, TLMs fail to copy the correct information from sequences of greater length than seen during training. Our hypothesis for this is that Transformers strongly rely on the position of the answer and have trouble learning simple tasks – such as copying the answer from the proof – if the information for this task happens at unseen positions.

**Q**: *Which reasoning strategy is easier to use when generating answers?* **A**: *backward-chaining is easier to use than forward-chaining and long-proofs are easier to use than short-proofs.*

Another interesting observation is that, in general, the reversed proofs (dotted lines in Figure 4) tend to be more useful than forward strategies for our model in generating the correct answer, aligning with our findings in Section 4.1. Similarly as above, we believe that this is due to the facts that Transformers strongly rely on the position of the answer. Indeed, in reversed proofs (**spr**, **lpr**), the answer is always in the first proof step, for which the position depends only on the story length; whereas in **sp** and **lp** the answer is always in the last proof step, for which the position depends both on the story length and on the proof length.

We also see that long, exhaustive proofs are easier to be used when generating the final answer, compared to short-proof strategies. This suggests that while being a longer sequence of tokens to encode, if a model was able to generate such proofs, it would ease its generalization capacities.

## 5 Conclusion

TLMs are state of the art models for a wide variety of natural language processing tasks. Given their widespread use, it is important to understand the limits of their ability to reason on knowledge expressed in natural language and to extrapolate learned inference procedures to unseen problem instances. Our explorations reveal multiple insights. Firstly, TLMs suffer from length-generalization issues in generating proofs. Secondly, TLMs get better at reasoning when trained with longer, exhaustive proofs. In addition, the fact that backward-chaining proof models perform better than forward-chaining ones makes us believe that backward-chaining strategies are easier to use albeit being harder to generate. Moreover, we find that no-proof models perform better than those trained to produce proofs. We conjecture that benefiting from naturally stated logical proof statements requires more complex internal representations. Recent work on developing position-agnostic attention mechanisms for Transformers (Dubois et al., 2020) can be useful as a future direction to develop generalizable models. Furthermore, our results motivates the use of neuro-symbolic methods such as Neural Theorem Provers (Rocktäschel and Riedel, 2017) as an alternative avenue to achieving systems that systematically generalize on logical and compositional reasoning tasks. Combining these approaches with large pre-trained language models is left as future research. We hope that this work will inspire research on the systematic generalization capacity of language models and motivate further study and the creation of neural models with greater reasoning capacity.

## Broader Impact

Transformer based models have been very effective for various language understanding and generation tasks. Due to their recent successes, there is significant interest in the applications of these models to real world scenarios such as: Dialogue, Question Answering and text-classification. However, failure of such systems could produce nonsensical, wrong or racially-biased results (Henderson et al., 2018). Therefore, a logical analysis of their limitations and issues in generalization to unseen data, such as in this work, could have a positive impact on building safer, more robust and interpretable systems in these domains.

In this work, we rely on systematic tests to trigger Transformer-based models to generate an interpretable proof in natural language, and then evaluate the robustness properties of that proof. Using a first-order logic based dataset, we explicitly test the logical consistency of such proof. This research can shed some light into developing more robust and systematic models in the future. In addition, it can help us understand the reasoning strategies employed by Transformer-based models for both inference and generation. However, the fact that proof-free inference works so well, may also imply that models which generate proofs, do so in a decoupled way from the computations yielding the final answer. This negative result could give users a false sense of explainability.

## Acknowledgments and Disclosure of Funding

The authors would like to acknowledge support from Element AI for providing computational resources which were used to run the experiments in this work. We also acknowledge the help provided by Sandeep Subramanian in sharing part of his experimental code. Nicolas is partially funded by a scholarship from the Fonds de Recherche Quebec Nature et Technologie. We thank CIFAR for their support through the CIFAR AI Chairs program. We also thank NSERC and PROMPT for their support.

## Footnotes

[1]Dataset and code can be downloaded at https://github.com/NicolasAG/SGinPG

[2]$k = 20$ in our case because we know that the maximum number of entities in a story is less than 20.

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
