[Supplementary Material]

# 6 Supplementary Material

## 6.1 Original CLUTRR evaluation

| ORIGINAL TEST | lvl.2 | lvl.3 | lvl.4 | lvl.5 | lvl.6 | lvl.7 | lvl.8 | lvl.9 | lvl.10 |
|---|---|---|---|---|---|---|---|---|---|
| **proofs** (*many proof steps*) | 99.62% | 0% | 0% | 0% | 0% | 0% | 0% | 0% | 0% |
| **proof steps** (*"since A-r1-B and B-r2-C then A-r3-C"*) | 99.62% | 0% | 99.96% | 0% | 100% | 0% | 0% | 0% | 0% |
| **facts** (*A-r-B*) | 100% | 0.47% | 100% | 0.83% | 100% | 0.20% | 0.20% | 0.10% | 0.42% |
| **entities** (*A*) | 100% | 23.81% | 100% | 35.72% | 100% | 26.19% | 21.43% | 30.95% | 30.95% |
| **relations** (*r*) | 100% | 100% | 100% | 100% | 100% | 100% | 100% | 100% | 100% |

Table 4: Percentage of the **original test** proof building blocks also present in the training set (composed of levels 2, 4, 6) for all levels. We colored all cells with a value close to 100% to better visualize which building blocks were entirely contained in the training set.

The original CLUTRR data generation framework made sure that each test `proof` is not in the training set in order to test whether a model is able to generalize to unseen proofs. Initial results on the original CLUTRR test sets resulted in strong model performance ($\sim 99\%$) on levels seen during training (2, 4, 6) but no generalization at all ($\sim 0\%$) to other levels. After further analysis, we noticed that due to the cloze style nature of CLUTRR tasks, the first names representing entities were chosen arbitrarily. This resulted in level-$k$ test set's `proof_steps` and `facts` also being in the level-$k$ training set. In addition, level-$k$ test set's `entities` were mostly seen *only* in level-$k$ training set. This resulted in a big overlap between training and test sets for examples of the same level, but a weak overlap on other levels as we can see in Table 4.

| NAMED TEST | lvl.2 | lvl.3 | lvl.4 | lvl.5 | lvl.6 | lvl.7 | lvl.8 | lvl.9 | lvl.10 |
|---|---|---|---|---|---|---|---|---|---|
| **proofs** (*many proof steps*) | 2.13% | 0% | 0% | 0% | 0% | 0% | 0% | 0% | 0% |
| **proof steps** (*"since A-r1-B and B-r2-C then A-r3-C"*) | 2.13% | 0% | 1.33% | 1.74% | 1.42% | 1.80% | 1.38% | 0.99% | 1.40% |
| **facts** (*A-r-B*) | 15.48% | 5.52% | 6.77% | 10.92% | 6.38% | 9.63% | 10.51% | 10.33% | 8.33% |
| **entities** (*A*) | 100% | 100% | 100% | 100% | 100% | 100% | 100% | 100% | 100% |
| **relations** (*r*) | 100% | 100% | 100% | 100% | 100% | 100% | 100% | 100% | 100% |

Table 5: Percentage of the **Named test** proof's building blocks also present in the training set (composed of levels 2, 4, 6) for all levels. We colored all cells with a value of 100% to better visualize which building blocks were entirely contained in the training set

In our case, the entity names are important to evaluate systematic generalization. We want to evaluate the capacity of a model to generalize to new `facts`, `proof_steps`, and `proofs`, but keeping the `entities` and `relations` the same. We thus modified the original CLUTRR dataset to select test entities according to entities present in the training set. We devise a test set that uses all `relations` and `entities` from the training set but new `facts`, `proof_steps` and `proofs` for all levels. We call this dataset the *Named* data: all entities are referred by their original first name. Train and test overlap percentages between all building blocks are in Table 5.

Given as input the story and the query followed by the proof trigger token ("<STORY> [story] <QUERY> [query] <PROOF>") the model generated the corresponding proof ans answer. We report

Figure 5: Answer accuracy on the Named test for all levels from 2 to 10. The models are given as input "<STORY> [story] <QUERY> [query] <PROOF>" and asked to generate the proof and answer. Models are trained on levels 2, 4, 6 only. Different proof settings are evaluated: **sp**=short-proof, **lp**=long-proof, **np**=no-proof. We also report the naive most-frequent-relation (**mfr**) baseline.

(a) Proof validity on the Named test for all levels from 2 to 10. The models are given as input "<STORY> [story] <QUERY> [query] <PROOF>" and asked to generate the proof and answer.

(b) Answer accuracy on the Named test for all levels from 2 to 10. The models are given as input "<STORY> [story] <QUERY> [query] <PROOF> [proof] <AN-SWER>" and asked to generate the answer.

Figure 6: Evaluation of models trained on levels 2, 4, 6 only.

in Figure 5 the answer accuracy and in Figure 6a the proof validity of all our models. Similarly, in Figure 6b we report the answer accuracy of our models when they are given as input the story, the query and the real proof, followed by the answer trigger token ("<STORY> [story] <QUERY> [query] <PROOF> [proof] <ANSWER>").

Experiments on this setup show that Transformer language models fail to generalize to unseen `facts`. Indeed, due to the presence of a large number of entities in CLUTRR, we end up with combinatorially large number of possible facts. The model may thus not be able to learn how to represent each entity effectively, hence reducing its chances to learn higher-order structures such as unseen `facts`.

## 6.2 Long Proof pseudo-code

```
def get_long_proof(story_facts, rules, query):
    """
    :params story_facts: list of (e_1, r, e_2) facts
    :params rules: list of composition rules. each rule is a dict
                   of the form {r1--r2: r3}
    :params query: tuple of entities for which we must find a relation (src, tgt)
    """
    proof = []  # list of proof steps to return

    # get all known relations (original, and reversed)
    all_facts = []
    for (e1, r, e2) in story_facts:
        inv_r = reverse_fact(e1, r, e2)
        all_facts.append((e1, r, e2))
        all_facts.append((e2, inv_r, e1))

    # go through every possible pair of facts
    for f1, f2 in itertools.combinations(all_facts, 2):
        e11, r1, e12 = f1
        e21, r2, e22 = f2
        inv_r1 = reverse_fact(e11, r1, e12)
        inv_r2 = reverse_fact(e21, r2, e22)

        # find the possible AB+BC combination.
        # there are 4 possible ways to combine 2 sentences with 2 entities each (1 in common):
        if e11 == e21 and e12 != e11 and e12 != e22:
            # AB+BC <=> inv_f1+f2
            A, new_r1, B = e12, inv_r1, e11
            B, new_r2, C = e21, r2, e22
            inv_r1 = r1
        elif e11 == e22 and e12 != e11 and e12 != e21:
            # AB+BC <=> f2+f1
```

```
            A, new_r1, B = e21, r2, e22
            B, new_r2, C = e11, r1, e12
            # swap inv_r1 and inv_r2
            inv_r1, inv_r2 = inv_r2, inv_r1
        elif e12 == e21 and e11 != e12 and e11 != e22:
            # AB+BC <=> f1+f2
            A, new_r1, B = e11, r1, e12
            B, new_r2, C = e21, r2, e22
        elif e12 == e22 and e11 != e12 and e11 != e21:
            # AB+BC <=> f1+inv_f2
            A, new_r1, B = e11, r1, e12
            B, new_r2, C = e22, inv_r2, e21
            inv_r2 = r2
        else:
            # invalid pair of facts
            continue

        # try to combine AB+BC
        if new_r1--new_r2 in rules:
            r3 = rules[new_r1--new_r2]
            inv_r3 = reverse_fact(A, r3, C)
            all_facts.append((A, r3, C))
            all_facts.append((C, inv_r3, A))
            proof.append(since A new_r1 B and B new_r2 C then A r3 C)
        # try to combine CB+BA
        elif inv_r2--inv_r1 in rules:
            r3 = rules[inv_r2--inv_r1]
            inv_r3 = reverse_fact(C, r3, A)
            all_facts.append((C, r3, A))
            all_facts.append((A, inv_r3, C))
            proof.append(since C inv_r2 B and B inv_r1 A then C r3 A)
        else:
            # invalid pair of facts
            continue

        # check if we found the link between the two queried entities
        (A, r, B) = all_facts[-1]
        if A==query[0] and B==query[1]:
            break
        if A==query[1] and B==query[0]:
            break

    return proof
```

## 6.3 Experiments parameter settings

| | small | large |
|---|---|---|
| patience | 20 | 20 |
| batch size | 512 | 256 |
| float precision | 16 | 16 |
| embedding dimension | 192 | 768 |
| number of layers | 5 | 20 |
| dropout | 0.1 | 0.1 |
| transformer mlp hidden size | 768 | 3072 |
| attention heads | 3 | 12 |
| max length | $1,024$ | 512 |
| activation | gelu | gelu |
| number of warmup steps | $20,000$ | $20,000$ |
| optimizer | adam | adam |
| **total parameters** | $\sim 3,000,000$ | $\sim 145,000,000$ |

Table 6: Parameter settings.

All experiments in the main section of the paper were run with the **small** model size.

Additional experiments in Section 6.4 were run with the **large** model size.

## 6.4 More parameters

In this section we report the answer accuracy of a model trained with ∼145M parameters and compare its generalization performance with our initial smaller network (∼2.5M parameters). Models are trained on levels 2, 4 and 6. Each model is given the story and query as input, and triggered to generate the proof and answer with the "<PROOF>" and "<ANSWER>" tokens respectively.

We observe in Figure 7 that the generalization capacity of the larger 145M network is almost identical to the smaller 2.5M parameter network trained on the same data (`facts` stories and short-proof-reversed). In addition, we also observe that the 145M model trained on reversed short proofs (145M / spr) is not better than the 2.5M model trained without any proof (2.5M / np). Overall, results show that model size improves only marginally the generalization capacity in our task.

Figure 7: Answer accuracy for all test levels from 2 to 10. The models are given as input "<STORY> [story] <QUERY> [query] <PROOF>" and they generate the proof and answer. Models are trained on levels 2, 4, 6 only. Stories are expressed with the `facts` template. Different proof settings are evaluated: **np**=no-proof and **spr**=short-proof-reversed. We also report the naive most-frequent-relation (**mfr**) baseline. Results on other proof settings with the 2.5M parameter network can be found in Figure 2a.

## 6.5 Fine-tuning GPT2

In this section we report the answer accuracy of GPT2 models (Radford et al., 2019) trained from-scratch (`gpt2FS-`) on the CLUTRR dataset and of pre-trained GPT2 models fine-tuned (`gpt2FT-`) on the CLUTRR dataset. We leverage the GPT2 implementation from the huggingface library (Wolf et al., 2019). The resulting models have ∼125M parameters. In all experiments the models are trained on stories expressed in the `amt` template. Models are fine-tuned on levels 2, 4 and 6. Each model is given the story and query as input, and triggered to generate the proof and answer with the "<PROOF>" and "<AN-SWER>" tokens respectively.

In Figure 8 we observe that in general, fine-tuned models perform better than the ones trained from scratch. We can

Figure 8: Answer accuracy for all test levels from 2 to 10. The models are given as input "<STORY> [story] <QUERY> [query] <PROOF>" and they generate the proof and answer. Models are fine-tuned on levels 2, 4, 6 only. Stories are expressed with the `amt` template. Different proof settings are evaluated: **sp**=short-proof, **spr**=short-proof-reversed, **np**=no-proof. We compare the performance of models trained from scratch (dotted lines; `gtp2FS-`) and of fine-tuned models (solid lines; `gpt2FT-`).

also see that reversed-proof strategies are better than their forward proof counterpart, which is in accordance with what we discussed in Section 4.1. Although fine-tuning seems to improve the generalization capacity of GPT2, it is also interesting to note that the benefit of fine-tuning GPT2 on short-proofs (sp) is negligible compared to the benefits of fine-tuning GPT2 on short-proofs-reversed (spr) or no-proof (np). This suggests that fine-tuning alone is not enough to yield strong generalization performance, but the choice of proof strategy also influences greatly the answer accuracy.

## 6.6 Encoder-Decoder Network

In this section we evaluate the answer accuracy of sequence-to-sequence models trained on `facts` templated stories of level 2, 4 and 6. These models consist of a 5-layer Transformer encoder and a 5-layer Transformer decoder, each of them following the same parameter settings than what is described in the '**small**' column of Table 6. This resulted in 5.22M parameter models. Sequence-to-sequence models are trained to encode the story and question with the encoder, and generate the proof and answer with the decoder. Models trained on levels 2, 4 and 6. Each model is given the story and query as input, and triggered to generate the proof and answer with the "<PROOF>" and "<ANSWER>" tokens respectively.

Figure 9: Answer accuracy for all test levels from 2 to 10. The models encodes the input "<STORY> [story] <QUERY> [query]" and they decode the proof and answer. Models are trained on levels 2, 4, 6 only. Stories are expressed with the `facts` template. Different proof settings are evaluated: **sp**=short-proof, **spr**=short-proof-reversed, **lp**=long-proof, **lpr**=long-proof-reversed, **np**=no-proof. We also report the naive most-frequent-relation (**mfr**) baseline.

In the results shown in Figure 9, we see that sequence-to-sequence models do not generalize well to unseen difficulty levels, both in extrapolation settings (levels 7–10) but also in interpolation settings (levels 3 and 5). This suggests that encoder-decoder architectures are more sensible to the sequence length seen during training. On the other hand, it is important to note that the encoder network was trained with the auto-regressive language modeling objective back-propagated from the decoder. It would be interesting to see if pre-training the encoder with a more traditional objective, that is masked language modeling (Devlin et al., 2019), would improve the generalization performance. We leave this exercise as future work. In addition, we plan to explore pre-trained models such as T5 (Raffel et al., 2020) in future work in order to improve performance with this type of architecture.