[Reviews · NeurIPS 2020]

Review 1

Summary and Contributions: Study of the ability of transformer language models (TLMs) to logically reason and generate templated proofs. Show that they fail to systematically generalize in various ways.

Strengths: Examining the systematic generalization and logical reasoning properties of TLMs is interesting, important and timely since these models have become very popular due to recent successes. Paper nicely examines these issues, and demonstrates that they fail to generalize in various desirable ways, particular with respect to generalizing to longer proofs than those seen during training. Illustrates some fundamental limitations of these popular and frequently exaggerated, over-estimated methods.

Weaknesses: Could include in discussion/conclusion that the inability of such models to systematically generalize is potential further motivation for neurosymbolic methods such as Neural Theorem Proving and TensorLog as an alternative avenue to achieving systems that systematically generalize for effective "soft" logical, compositional, complex reasoning. How would one directly experimentally compare to these methods on the studied problems? Did not explore position-agnostic attention, which is left for future work.

Correctness: Work seems technically solid and well conducted. Experimental method seems sound and reasonable.

Clarity: Paper is fairly well written and clear. The different uses of NL proofs versus triple-based versions is a bit confusing at times. Could be clearer about exactly when and where these two different representations are used as either input or output formats. Minor: The F. Petroni reference is messed up, this is not the first author. I wish folks would proofread their bib better.

Relation to Prior Work: Good coverage of related work to my knowledge. Could cite work on neurosymbolic methods (e.g. neural theorem proving) methods as an alternative to trying to get purely neural models to achieving "soft" compositional, complex reasoning.

Reproducibility: Yes

Additional Feedback: Would a graph representation of the input processed by a Graph Transformer model (https://arxiv.org/abs/1911.06455) potentially generalize better by exploiting the structured relational representation in the input and output


Review 2

Summary and Contributions: This paper studies the ability of transformer language models to generate proofs by chaining known facts to produce new ones, and thereby answer queries about the entities involved. The paper shows both positive and negative results in this setting by constructing training and test data from the CLUTRR dataset. The paper shows that these models generalize better when no proofs are involved. When a proof is required, they don't generalize to proofs that are longer than what they saw at training time, but do generalize to shorter ones. Moreover, the results show that the models prefer backward chaining proofs more. *Update* (after author feedback): Thanks for your responses. I would strongly encourage that the natural language experiments be moved from the appendix to the main body, along with discussion that highlights the strengths and potential limits of the strategy explored in this work. (Much of this could come from the reviews and the feedback.)

Strengths: The paper makes a focused contribution about a specific question and the experiments show interesting results. Moreover, the fact that not all results are positive opens up questions for future research about why, and also may spur potential modeling innovations that address these issues.

Weaknesses: While the experiments are interesting, the paper seems to oversell its claims. None of the experiments in the main body of the paper involve natural language because the "sentences" are produced with what seem like a fixed template. Natural language becomes much more complex due to diversity at the lexical and syntactic levels in expressing the same ideas. In contrast, the sentences in this paper are all the same template. In essence, it is presenting a sequence of triples to the transformer model and asking it to generate the more triples and a relation as the answer. The other words in the templates are basically separators that are devoid of meaning as far as this study is concerned. Hence my concern raised above. On actual natural language, as seen from the appendix, the conclusions are weaker. (Please correct me if this is incorrect. The above points don't make the paper less interesting. However, as I understand it, the paper is misleading.)

Correctness: The empirical methodology and the claims seem to be fine, but for the weaknesses described above.

Clarity: The paper is very well written.

Relation to Prior Work: Yes.

Reproducibility: Yes

Additional Feedback: What the paper is evaluating is the ability of the models to chain information systematically. For this goal, is the transformer language model the right object of study? A big part of the impressive performance that these models offer comes from the fact that they are trained on massive *natural* textual corpora. Clarification question: The paper says in line 181 that the model is being trained on all entities, relations and facts. If the training involves all possible facts, then shouldn't all proofs be just stating the fact (because it is already known)? That is, if the query is something like ("Anna", _, "Bob") and the relation is known to the model at training time, then why can't it just produce it directly in one step? The supplementary material says that the number of layers is 5. Does adding more layers improve proof length generalization? How sensitive are the conclusions of the paper if there are more or less transformer layers? Would the conclusions change if a BERT-family of models were used instead of the simpler transformer here? (For example, the work of Clark et al 2020 that is cited here shows results with RoBERTa.) The COMET work from AI2/Univ of Washington seems somewhat related here in that it also uses a transformer model to produce commonsense inferences. Minor: * The definition of difficulty of the tasks using the number of edges in the graph may be valid in this data, but only seems to be partially correct. The complexity of the proof is the number of steps in the proof, rather than the graph. The graph may have additional edges that are not involved in the proof. If this is not the case, then the problems considered here are more simple than the general case. * Line 133: Section 3.2 -> Section 3.1


Review 3

Summary and Contributions: This paper investigates the systematic generalization abilities of transformer-based language models through their performance on an inductive logical reasoning task in natural language. The paper test the proof generation capabilities and the inference capabilities of transformer language models and highlights some observations that maybe useful for investigating the reasoning ability and strategies of transformer language models. == posted after author feedback == I've read the author response and other reviews. Some of my concerns are addressed in their response, namely the part with larger transformer models or pre-trained LMs. My remain concern is the relation between the employed "toy" task and NLG tasks that require more complex reasoning. That's to say, the conclusion drawn in the paper may be too closely related to the proof generation task rather than to the systematic reasoning ability of transformer models. However, given that the paper is well written and provides some insights, I am OK if other reviewers would like to see it in the program. I have increased the score from 4 to 5.

Strengths: The problem investigated by this paper is interesting and important. The experimental settings are clearly described and the results are thoroughly analyzed.

Weaknesses: The paper only conduct experiments on one datasets, which limits the generalizability of the empirical findings. In addition, the dataset used in the paper is somewhat toy (only 90 tokens and limited relations). Also, the transformer model size described in the paper contains only 2.5M parameters, which is much smaller than typical transformer language models that contain 100M parameters or more. It would be also interesting to investigate the performance of pre-trained language models like gpt-2 on the evaluated settings. Finally, the findings in the paper (e.g. length-generalization issues, which is common in the NLG literature) are not surprising and are not sufficiently significant to convey much insights for future works. For example, the backward-chaining strategy for proof generation is easier to use because the target is generated at first, which seems straightforward. Also, the difference between joint learning the proof generation model and the inference model seems not very significant.

Correctness: The empirical methods are correct and the claims seem to be reasonable for me.

Clarity: The paper is well written and easy to follow.

Relation to Prior Work: The relation to prior work is clearly discussed.

Reproducibility: Yes

Additional Feedback: The suggestions can be found in the weakness part of the review.


Review 4

Summary and Contributions: This paper evaluates how well Transformer language models can generate natural language expressions corresponding to first-order logical proofs, and their answers. Given a dataset of facts (tuples like entity1-relation1-entity2, entity2-relation2-entity3) and a query (entity1-?-entity3), the language model is trained on a sentence representing the facts, the query, a proof, and the answer. The proof is a chain of implications (for example, one step is "since entity1 is in relation1 with entity2 and entity2 is in relation2 with entity3, then entity1 is in relation2 with entity3"). The answer is the missing relation, such as relation2. The model can then be tested by presenting only the prefix of the expressions corresponding to the facts and the query (and perhaps the proof), and predicting the answer. The paper evaluates the ability of Transformer language models to generalize in several settings, determined by the number of relations. The main findings are that these models generalize well to lengths similar to those seen in training, but not to longer lengths. The models also work better on backward-chaining proofs, meaning when the sought relation is in the first expression, and subsequent expressions contain the proof steps required to arrive at it. Finally, a model without the proof tends to work better at arriving at the answer than models with proofs. The main contributions of this paper are the formulation of logical theorem proving as language expressions that are amenable to be processed by language models, and the evaluation of Transformer language models in several settings. == edit after author response == Thank you for your detailed response. You've answered most of my comments and I'm glad to already see some results with larger models. (Adding an RNN experiment could also be interesting, if you have time and space.) Hopefully, the comments will help rephrase some of the points, and improve the introduction, to make the paper well motivated and the results analyzed in more detail. One lingering issue us the no-proof setup working so well (see last point in weaknesses). =========================

Strengths: 1. The paper targets an important problem, namely systematic generalization, and evaluates a commonly used model in the NLP community, the Transformer. 2. Its formulation of the problem and the empirical evaluation is novel as far as I am aware. 3. The experiments are mostly well planned, comparing generalization in seen and unseen proof lengths. 4. The questions are of interest to the machine learning / AI community, although not motivated enough (see below).

Weaknesses: 1. The paper motivates the investigation via systematic generalization, but there is insufficient motivation for studying theorem proving. I agree it's an important problem to study, but why should language models be expected to perform it? Some of it is addressed in the broader impact section, but I would use the introduction to motivate the problem more and explain why *natural* language models should do it. How do the questions relate to the way language models are typically used? 2. There is very little discussion of the literature on theorem proving. Given that this has been a major topic of AI research in the past, more discussion would be useful. 3. The paper studies only one kind of neural architecture, Transformers, and although these are very popular recently, I wonder if the conclusions would hold in other architectures. In particular, it is known that the Transformer is especially helpful with long distance relationships, but the present paper uses relatively short examples, so maybe other models would be just as good. Or maybe other models would suffer from different limitations. 4. The initial experiments show that the models fail to generalize to new facts and thus most of the experimental evaluation is made in a simpler setup, where the model is exposed to all facts (the entity-relation-entity tuples), and needs to generate new proof steps ("since entity1 is in relation1 with entity2 and entity2 is in relation2 with entity3, then entity1 is in relation2 with entity3"). While I appreciate the honesty of disclosing this limitation, I'm afraid it's a major one. In fact, the correct answer is seen in the prefix presented to the model, and it then needs to generate it. This may also explain why the backward chaining works so much better: the model just needs to know to look for the fact in the first sentence (proof step), which makes this a kind of memorization task. Then again, even with forward chaining, the model should only need to locate the last sentence and extract the answer out of it. Why is this not easy for the model to do? 5. Relatedly, the fact that the no-proof setup works so much better is really worrying. This isn't strictly a weakness. The paper justly highlights this in section 4.3. But, I would like to see more analysis of this point and in particular an analysis of how the models arrive at their decisions. Do they do any reasoning? Do they just learn to focus on the first/last proof step and extract the answer? What does the no-proof model do? For example, tracking the attention behavior or using some saliency maps could provide some insights on these questions.

Correctness: The claims are mostly correct based on the experiments carried out. The methodology seems reasonable, except for the limitation of providing all facts to the models, discussed above.

Clarity: The paper is clearly written. I spotted occasional typos or minor grammar or formatting issues: - line 326: "is" -> "are" - line 133: Section 3.2 -> 3.1 - line 72: "There have been several recent research (studies?)" - line 78: SCAN dataset... have... - And a few others

Relation to Prior Work: - More discussion of theorem proving in AI would be helpful. - More discussion of using neural networks for theorem proving. A bit is mentioned in the intro, but this can be expanded.

Reproducibility: Yes

Additional Feedback: 1. Have you considered initializing the models with pre-trained models? Would that affect their reasoning abilities? Or even just evaluate a pre-trained model without further fine-tuning it. 2. How exactly could this work "inspire future research directions for the community to design models with greater reasoning capacity"? The conclusion mentions the position-agnostic attention. Any more concrete thoughts? 3. How "natural" are the language expressions? How diverse are the templates and their instantiations? The amt experiments in 6.2 show that more natural language leads to much worse generalization. I think it's important to highlight this point in the main paper. 4. What is k in section 3.3? 5. Section 4.2 seems to be in contradiction to 4.1: backward chaining strategies lead to better answer accuracy but worse proof validity. How can this be reconciled? To me, it seems like in the backward-chaining strategy all that matters is getting the first step right (or even just have the correct fact in the beginning), and then the model can extract the answer. Can this be analyzed?

[Author Response · NeurIPS 2020]

We thank the reviewers for their detailed comments and their useful suggestions. We are excited that **R1** and **R4** find our work interesting, timely and novel, and that our results demonstrate the fundamental limitations of Transformer Language Models (TLMs) reasoning abilities. We thank **R2** and **R3** for acknowledging that our experiments show interesting results that open up questions for future research and spur potential modeling innovations. Some of the concerns seen in the reviews were common to more than one reviewer, so we address these by topic below.

@**R1**, **R2**, **R3**, **R4** **Generalization to larger networks and different architectures**. We thank the reviewers for suggesting to improve the experimental results by analyzing different model architectures, such as larger transformer models (**R2**,**R3**), pre-trained language models (**R3**,**R4**), Transformer encoder-decoder (**R2**) and Graph Transformers (**R1**). In this rebuttal, we report results on larger transformer models. We agree to provide results of pre-trained models like GPT-2 and Transformer encoder-decoder models in the final submission, and if time permits Graph Transformers. Regarding training using a larger Transformer model (**R2**,**R3**), we agree that 2.5M parameters is small compared to more traditional transformer architectures such as GPT-X. To address this concern, we trained a 20 layer auto-regressive network, resulting in 145M parameters. We observe that the generalization capacity of this network is similar (43%) to the 2.5M parameter network trained on the same data (46%). Our preliminary investigation on pre-trained language models (**R3**,**R4**), suggests that the pre-trained model has similar trends as training from scratch but we acknowledge it needs further investigation. Due to specific limitations during training, preliminary investigation on Transformer encoder-decoder (**R2**) suggests weaker generalization scores which we aim to investigate further.

@**R1**, **R2**, **R3**, **R4** **On the motivation for using TLMs**. This is an excellent question that we think will benefit the understanding of all reviewers. We agree with **R2** that existing literature explores pre-training abilities of TLMs on large natural language corpora. While training on massive data can give certain advantages with respect to understanding the meanings of words, we conjecture that such data gives models much less experience with reasoning over long inference chains. We study the less understood issues related to how well TLMs are able to perform long chains of reasoning. Moreover, recent work such as LAMA, T5 and GPT3 suggest that language models can be treated as knowledge bases. This directly motivates us to investigate if language models can also learn certain reasoning strategies. Studying these abilities would enable future research in using these models as dynamic knowledge bases that could infer new knowledge even when it is not "stored" directly (i.e. seen during pre-training). We will add this discussion to the paper.

@**R2**, **R3** **Natural Language results**. We thank you for highlighting the importance of results on natural language stories. We acknowledge that generalization is weaker in this harder setting, and we confirmed that by performing additional experiments on the natural language split. We still find that the proof resolution strategy influences the generalization capacity of TLMs. In particular, the conclusion that models trained on long, exhaustive proofs generalize better than short proofs still holds. We plan on moving discussion from the Appendix to the main section of the paper along with additional results.

@**R2**, **R3** **On the complexity of the task**. We acknowledge that the dataset in use (CLUTRR) is a toy dataset. However, this in turn allows us to carefully analyze and control the difficulty of the experiments. For instance, with 20 possible entities ($k$ in Section 3.3) and 20 possible family kinship relationships, the model have to learn 8,000 possible triples. Given that even in this simplistic setup the generalization performance is not positive, this warrants a deeper inspection of reasoning mechanisms of TLMs. We plan on extending our experiments with other datasets in the future.

@**R2**, **R4** **On the issue of "all facts are seen" / "the correct answer is seen in the prefix"**. While our models have seen all possible facts in all training proofs, the target answer to a (story, question) pair is not seen in the prefix given to the model, unless the proof is explicitly given as in Section 4.3. Here, our experiments reveal that beyond 7-step proofs, the copy mechanism learned by TLMs becomes unreliable due to positional token embeddings. Position-agnostic embeddings could help in solving this issue, which we leave as an exercise for future work.

@**R4** **On the backward-chaining contradiction**. It is a great question why backward-chaining proofs are easier to use in Section 4.1, but are harder to generate in Section 4.2. We also agree with **R4**, this is due to the fact that backward chaining proofs contain the answer in the first proof step. Thus, there is a higher probability of the model to generate this step correctly and then use it while predicting the answer. This explains why the answer accuracy of such model is relatively high while their proof validity is low. We will note this phenomenon in the final version of the paper.

In general, we will fix typos and broken references (**R1**,**R2**,**R4**), clarify the presentation and some notations (**R1**,**R4**), and expand on the background section by discussing theorem proving (**R1**,**R4**). We would like to thank again all the reviewers for their time and effort in reading our paper and giving us good feedback and suggestions.

[Meta-Review · NeurIPS 2020]

This paper evaluates a trained-from-scratch Transformer language model on an artificial simple-theorem-proving task in a way that helps to highlight and clarify some limitations of this commonly-used architecture. Reviewers found some points in the motivation and in the discussion of results potentially a bit misleading, especially surrounding the connection between this work and natural language, but ultimately formed a consensus that the primary claims of the paper are sound and significant, and that the remaining presentational issues don't undermine that.